# Anti-Toxin Responses to Natural Enterotoxigenic *Escherichia coli* (ETEC) Infection in Adults and Children in Bangladesh

**DOI:** 10.3390/microorganisms11102524

**Published:** 2023-10-09

**Authors:** Petra Girardi, Taufiqur Rahman Bhuiyan, Samuel B. Lundin, Shushan Harutyunyan, Irene Neuhauser, Farhana Khanam, Gábor Nagy, Valéria Szijártó, Tamás Henics, Eszter Nagy, Ali M. Harandi, Firdausi Qadri

**Affiliations:** 1Eveliqure Biotechnologies GmbH, Karl-Farkas-Gasse 22, 1030 Vienna, Austria; shushan.harutyunyan@eveliqure.com (S.H.); irene.neuhauser@eveliqure.com (I.N.); tamas.henics@eveliqure.com (T.H.); eszter.nagy@eveliqure.com (E.N.); 2Infectious Diseases Division, International Centre for Diarrhoeal Disease Research Bangladesh (ICDDR,B), 68 Shaheed Tajuddin Ahmed Sarani, Mohakhali, Dhaka 1212, Bangladesh; taufiqur@icddrb.org (T.R.B.); farhanak@icddrb.org (F.K.); fqadri@icddrb.org (F.Q.); 3Department of Microbiology and Immunology, University of Gothenburg, 405 30 Goteborg, Sweden; samuel.lundin@microbio.gu.se (S.B.L.); ali.harandi@microbio.gu.se (A.M.H.); 4CEBINA GmbH, Karl-Farkas-Gasse 22, 1030 Vienna, Austria; gabor.nagy@eveliqure.com (G.N.); valeria.szijarto@eveliqure.com (V.S.); 5Vaccine Evaluation Center, BC Children’s Hospital Research Institute, University of British Columbia, Vancouver, BC V5Z 4H4, Canada

**Keywords:** ETEC, heat-labile toxin (LT), heat-stable toxin (ST), natural infection, endemic population, children

## Abstract

A sero-epidemiology study was conducted in Dhaka, Bangladesh between January 2020 and February 2021 to assess the immune responses to ETEC infection in adults and children. (1) Background: Enterotoxigenic *Escherichia coli* infection is a main cause of diarrheal disease in endemic countries. The characterization of the immune responses evoked by natural infection can guide vaccine development efforts. (2) Methods: A total of 617 adult and 480 pediatric diarrheal patients were screened, and 43 adults and 46 children (below 5 years of age) with an acute ETEC infection completed the study. The plasma samples were analyzed for antibody responses against the ETEC toxins. (3) Results: Heat-stable toxin (ST)-positive ETEC is the main cause of ETEC infection in adults, unlike in children in an endemic setting. We detected very low levels of anti-ST antibodies, and no ST-neutralizing activity. However, infection with ETEC strains expressing the heat-labile toxin (LT) induced systemic antibody responses in less than 25% of subjects. The antibody levels against LTA and LTB, as well as cholera toxin (CT), correlated well. The anti-LT antibodies were shown to have LT- and CT- neutralizing activity. The antibody reactivity against linear LT epitopes did not correlate with toxin-neutralizing activity. (4) Conclusions: Unlike LT, ST is a poor antigen and even adults have low anti-ST antibody levels that do not allow for the detection of toxin-neutralizing activity.

## 1. Introduction

Despite the improvements over the last few decades in drinking water quality, sanitation, and the implementation of current prevention and treatment interventions, diarrheal diseases remain a major cause of illness, death, and developmental disabilities such as stunting and cognitive development especially among children in low- and middle- income countries (LMICs) under five years of age [1,2,3,4]. Severe watery diarrhea due to enterotoxigenic *Escherichia coli* (ETEC) represents a major cause of diarrhea among children <5 years living in endemic countries and also travelers to endemic regions.

In addition to the use of proven measures, a comprehensive approach to address diarrhea is urgently required, including the development of innovative tools to fill current gaps in effective prevention strategies. Vaccines in particular are a major tool for the prevention of diarrheal diseases [5,6] but currently the only enteric diseases/pathogens for which licensed vaccines exist are rotavirus, cholera, and *Salmonella* Typhi (against typhoid). 

ETEC pathogenesis is dependent on the production of diarrheagenic toxins: the heat-labile toxin (LT) and/or the heat-stable toxin (ST). LT and ST cause alterations in the ion and fluid transport of epithelial cells, which results in increased water secretion into the gut (diarrhea). This can contribute to severe dehydration with detrimental consequences especially in young children living in countries without an appropriate healthcare infrastructure. 

LT is a multi-component bacterial toxin assembled from two separately expressed subunits [7]. Following ETEC colonization and the subsequent toxin release in the small intestine, LTB subunits irreversibly bind GM1 gangliosides (a sialic acid-containing oligosaccharide covalently attached to a ceramide lipid) on the surface of epithelial cells [8]. LT is internalized and activates adenylate cyclase resulting in an increase in the intracellular cAMP level that leads to activation of the chloride channel. As a consequence, chloride ions and water are transported to the gut lumen. When this exceeds the absorption capacity, watery diarrhea occurs [8,9]. LT is highly immunogenic, and has been considered as a mucosal adjuvant. Since LT is analogous to the cholera toxin (CT) structurally, antigenically, and functionally, CTB has been included in cholera vaccines and is known to induce cross-reactive antibodies against LTB (Dukoral) that are suggested to confer protection against ETEC infections as well, which would be limited to the LT-expressing ETEC strains [10].

ST is an 18 or 19 amino acid-long cyclic peptide (STh and STp, respectively) that binds to its receptor, guanylate cyclase C (GC-C). This activation coincides with elevated cGMP production and leads to increased water retention in the gut (diarrhea). ST is a challenging vaccine antigen due to its small size and the consequent low immunogenicity, as well as its complex tertiary structure [8].

ETEC-induced diarrhea is mainly attributed to the effects of the LT and ST toxins. The antibodies neutralizing these ETEC toxins can be considered protective, although it is not known what levels of such antibodies are sufficient. Adults living in endemic regions acquire protective immunity due to repeated exposure, while young children are the most susceptible. We herein characterized the anti-toxin antibody profiles in plasma samples of infected adults and children living in Bangladesh, which is endemic for ETEC infections. Our goal was to assess the toxin-neutralizing potency of plasma from the acute infection phase and convalescence. This information is expected to guide vaccine efforts and approaches against ETEC infection.

## 2. Materials and Methods

### 2.1. Study Design

This study was approved by the Ethical Review Committee (ERC) of the icddr,b and conducted using Good Clinical Practice (GCP) and in accordance with the World Medical Association (WMA) Declaration of Helsinki. For screening diarrheal patients, the medical team identified and selected potential patients with typical symptoms of diarrhea from the selected health facilities in Mirpur as well as icddr,b Dhaka hospital. In general, patients with watery stool were examined as suspected cases of ETEC diarrhea. Stool samples from these patients were collected to detect ETEC by culture followed by PCR. A signed and dated informed consent document was obtained from adult participants or parents of child participants before initiating any study-specific procedure.

Inclusion criteria: 1. A positive stool culture for enterotoxigenic *Escherichia coli* and absence of significant comorbid conditions. 2. Male or female adult (18–45 years) and children (up to 5 years) patients. 3. Free of concurrent infections. 4. Adult participants or father/mother of child participants properly informed about the study, able to understand it and sign the informed consent form. 5. Available for the entire period of the study and reachable by study staff throughout the entire follow-up period.

Exclusion criteria: 1. History of immunization with experimental ETEC or Shigella vaccines. 2. Pregnant women. 3. Presence of a known significant medical or psychiatric condition that in the opinion of the investigator precludes participation in the study. 4. Those receiving immunosuppressive therapy. 5. Prior receipt of a blood transfusion or blood products, including immunoglobulins. 6. Evidence of current illicit drug use or drug dependence (adult participants only). 7. Any condition which in the opinion of the investigators puts the subject at risk of noncompliance with the protocol. 

### 2.2. Sample Collection

Blood samples from patients were collected at enrolment (day 2/3 after onset of diarrhea), and follow-up visits at day 7 ± 1 and day 30 ± 4 and once from healthy adult participants. Approximately 10 mL of blood was collected from adults and 5 mL from children. Blood samples were collected in a heparinized tube, and immediately transported to the icddr,b laboratory, maintaining cold chain. Plasma samples were separated from blood and stored in aliquots at −80 °C.

### 2.3. ELISA

ELISA plates with 96 wells or Streptawell ELISA plates were coated with the respective antigen: heat-stable toxin (ST, biotinylated synthetic peptide, PepScan, Lelystad, The Netherlands), heat-labile toxin B subunit (LTB, Sigma-Aldrich, Schnelldorf, Germany), heat-labile toxin A subunit (LTA, Szabo-Scandic, Vienna, Austria), or Cholera toxin B subunit (CTB, Sigma-Aldrich). Serial dilutions of plasma samples were run together with negative controls on each plate. Goat F(ab’)2 anti-human IgG-HRP (SouthernBiotech, Birmingham, AL, USA) or Goat Anti-Human IgA-HRP (SouthernBiotech) was used and detected either with Sureblue TMB (Seracare Life Sciences, Inc., Milford, MA, USA), blocked with H_2_SO_4,_ and OD_450nm_ was measured with an EON plate reader (BioTek, Winooski, VT, USA). Alternatively, ABTS substrate (Fisher Scientific, Vienna, Austria) was used and OD_405nm_ was measured with a Synergy HTX (BioTek) plate reader. Human serum samples from non-endemic healthy adults without any history of an ETEC infection were used as a negative control. The determined end point titer corresponds to the highest dilution factor giving a signal value above the mean signal of the negative control multiplied by the mean signal of the sample at the corresponding dilution. Alternatively, titer was determined using Gen5 (version number 2.01.14, BioTek) software.

### 2.4. Toxin Neutralization

Neutralizing capacity of anti-LTB or anti-CTB antibodies in serum was assessed in GM1 binding assay. Serum dilutions were incubated with 0.5 ng LTB or CTB (Sigma-Aldrich), and the amount of LTB or CTB that remained free from antibody binding was quantified by binding to GM1-coated plates by ELISA using an anti-cholera toxin beta antibody (Sigma-Aldrich) for detection. LT or ST neutralization was assessed by blocking of LT-induced cAMP or ST-induced cGMP induction in T84 human colon epithelial cells as previously described [11,12]. Briefly, T84 cells (ATCC) were seeded in 24-well plates in DMEM/F12 (Fisher Scientific) supplemented with 5% FCS (Sigma-Aldrich), P/S (Fisher Scientific) and grown until confluency. Cells were washed with medium (DMEM/F12 without FCS, P/S) and pre-incubated with 1 mM 3-3-isobutyl-1-methylxanthine (IBMX, Sigma-Aldrich). A total of 0.5 ng of LT or 1 ng and 5 ng synthetic ST (PepScan) was pre-incubated with selected human plasma samples with high LT or high ST titers correspondingly. As controls, serum of mice after 3× i.p. vaccination with 200 µg recombinant LTB-ST_N12S_ protein [12] was used for ST-neutralization assay and in-house human serum with high LT-neutralization capacity for LT neutralization assay. After pre-incubation of toxin and plasma/serum, the mix was transferred to T84 cells and incubated at 37 °C (5% CO_2_) for 3 h or 1 h for LT- or ST-assays correspondingly. Supernatants were removed, and cells lysed with 0.1 M HCl/1% Triton X-100 at RT. Cell lysates were centrifuged and supernatants were assessed for LT-induced cAMP or ST-induced cGMP by using direct cAMP/cGMP ELISA kits (Enzo Life Sciences, Lausen, Switzerland) according to manufacturer’s instructions. LT neutralization activity was calculated by multiplying the plasma/serum dilution factor with the percentage of LT neutralization at the corresponding plasma/serum dilution used. 

### 2.5. Linear Peptide Epitope Mapping

IgG antibody responses to LTB peptides were assayed using peptide array analysis. Medium-density arrays were created using laser jet-assisted on-chip synthesis technology. On these array chips, 93 12-amino acid (12-mer) overlapping LTA peptides and 74 12-mer LTB peptides were spotted onto each chip; the peptide location on the chip was randomized to reduce the influence of any spatial bias of the chip. Peptide sequences were of the LT1 variant (GenBank accessions CBJ04426.1 and CBJ04425.1 for LTA and LTB sequences, respectively), and also included peptide sequence variants of both LTA and LTB from LT2, LT3, LT4, and LT6 variants (GenBank accession numbers EU113246.1, EU113242.1, EU113243.1 and EU113245.1, respectively). The peptide sequences selected were sequential and overlapping and spanned the entire amino acid sequences of LTA and LTB with a sequence overlap of 10 amino acids between each peptide. In addition to LTA and LTB peptides, control peptides were also included on the array (sequences YPYDVPDYAG of Influenza hemagglutinin and KEVPALTAVETGAT of the capsid protein of Human poliovirus 2, in addition to the truncated “single-amino acid-peptide” G). All peptides were printed on 16 separate sectors of each array chip, allowing analysis of 16 different samples on each array. To map antibody binding to each peptide, each serum sample was incubated in a 1/500-dilution to a peptide array sector containing all LTA and LTB peptides and control peptides, followed by washing and subsequent incubation by DyLight680-conjugated goat anti-human-IgG (Fc) antibody. Finally, fluorescence image scanning using a Innopsys InnoScan 710-IR microarray scanner, and subsequent digital image analysis was performed to detect antibody binding to each of the peptides on the chip. Chip printing and antibody analysis was performed by PEPperPRINT (Heidelberg, Germany). Data for antibody binding to each peptide was generated by subtracting the assay background (fluorescence signal outside the peptide spots of the array) from the foreground signal (fluorescence signal inside the peptide spots of the array), for each sample. The cutoff for a positive signal was calculated as being 3 standard deviations above the average log fluorescence signal of the negative control peptide spots; this cutoff was 480 for IgG-responses.

## 3. Results

### 3.1. Study Participants

The study was conducted between January 2020 and February 2021 in Bangladesh in Mirpur, located in the Dhaka Metropolitan area. This urban research area has been subjected to field studies since 1987 and is densely populated with approximately 3.5 million individuals. Approximately 98% of the residents have access to tap water supplied by the municipality while the remainder use wells, hand pumps, and other sources such as ponds and rivers in the study area. As part of this study, participants were recruited from selected local health facilities in Mirpur as well as the icddr,b Dhaka hospital, presenting with clinical symptoms of diarrhea. A total of 1097 diarrheal patients (617 adults, 480 children) presenting with watery stool were screened for the type of infection by culture and PCR of the stool samples. One hundred-thirty ETEC-infected patients were identified and 97 met the inclusion/exclusion criteria and were enrolled in the study. Among the enrolled ETEC-positive adult patients, 43 completed the follow-up visits and seven were lost to the follow-up. In the pediatric group, 46 completed the follow-up visits and one was lost to the follow-up. The age and gender distribution of enrolled participants are detailed in Table 1. Among the enrolled children, 58.7% were between 2–5 years of age, while 10.9% and 30.4% were under the age of 1 year and 1–2 years old, respectively (Table 1). 

The ETEC infection was typed in the enrolled patients regarding the presence of the LT and ST genes (Table 2). Of the 46 enrolled children, 41.3% had an infection with LT-only positive ETEC. In contrast, among the 43 enrolled adults, only 14% had an ETEC infection caused by an ETEC strain expressing only LT. The two types of ST genes, STh and STp were distinguished.

### 3.2. Immune Responses against LT

The plasma from all the enrolled ETEC-infected patients was assessed for LTB-reactive IgG and IgA antibodies in acute, early, and late convalescence phases (within 2–3 days, day 7, and day 30 after presenting with symptoms, respectively). The individual titers are shown in Figure 1, and the responder rates are listed in Appendix A. Of the 43 adults and 46 children with confirmed ETEC infections, 28 (65%) and 34 (74%) had an infection with LT-expressing ETEC strains, respectively. Of these, only five (18%) adults and eight (24%) children had an at least 2-fold increase and four (12%) adults, and three (11%) children had an at least 4-fold increase in anti-LTB IgG plasma titer at any sampling day compared to day 2. For anti-LTB IgA, only three (11%) adults and seven (21%) children had an at least 2-fold increase and one (4%) adult, and one (3%) child had an at least 4-fold increase in plasma titer at any sampling day compared to day 30.

There was no correlation between the IgG and IgA antibody titers against LTB in adults, while in children a strong correlation was observed after infection with an LT+ ETEC strain (Figure 2). 

In addition to the LTB responses, we assessed the anti-LTA IgG antibody responses in all the ETEC infected adults and children enrolled in the study (individual levels shown in Figure 3). For anti-LTA IgG, five (18%) adults and six (18%) children had an at least 2-fold increase and 3 (11%) adults, and 3 (9%) children had an at least 4-fold increase at any sampling day compared to day 2. 

When we compared the anti-LTB IgG and ant-LTA IgG titers in children and adults, we observed that the IgG antibody responses against LTA and LTB correlated strongly in adults (Figure 4a), but weaker in children (Figure 4b).

The plasma samples from 23 selected subjects (including those of all subjects with a detectable increase in LTB titers and subjects with pre-existing high anti-LTB IgG titers, Appendix A) were analyzed for LT-neutralizing capacity using GM1 binding assay. For these subjects, the plasma samples from all three sampling times were assessed and the dilution factor for each individual plasma to achieve 50% blocking of the LTB binding to GM1 (DF50) was calculated. The DF50 value showed a very strongly positive correlation with the anti-LTB IgG titer of the individual samples (Figure 5a).

The same samples were assessed for anti-CTB IgG, and the anti-CTB IgG titer correlated strongly with the anti-LTB IgG titer (Figure 5b), which is not surprising given the high homology between LTB and CTB. Similarly to what was observed with LTB, the anti-CTB IgG titer also correlated strongly with the CTB neutralizing capacity (Figure 5c) as measured by blocking the binding of CTB to GM1. Furthermore, there was a strongly positive correlation between the LTB and CTB neutralization activities (Figure 5d).

The LT neutralization of the selected plasma samples was also assessed using a cell-based assay, measuring the LT-induced cAMP release from T84 human colon epithelial cells. The percentage of LT neutralization for the plasma/serum samples was calculated relative to the cAMP release level when LT without serum/plasma was used. The LT neutralization was calculated by multiplying the dilution factor with the neutralization percentage at the corresponding plasma dilution. The plasma samples from endemic, ETEC-infected subjects displayed a higher neutralizing activity than the control samples collected from non-endemic adult individuals, and the neutralizing activity increased in response to ETEC infection (Figure 6a).

We found a positive correlation between the LTB neutralizing capacity based on GM1 binding and inhibition of cAMP induction (Figure 6b).

The plasma samples from the selected individuals diagnosed with LT-positive ETEC infection and having high plasma IgG titers against LTB and/or increases in the plasma anti-LTB IgG titers were chosen for linear LTB epitope mapping. Epitope mapping was performed for the peptides obtained from all known LTB protein sequence variants (LT1, LT2, LT3, LT4, and LT6). Notably, only a few individuals had high reactivity against the 12-mer peptides and even fewer samples showed reactivity to overlapping peptides (Figure 7). In these few cases, the epitope reactivity did overlap with previously published LTB epitopes (indicated in Figure 7) [13,14]. The same observation was made with the peptides covering LTA (Figure 8) [15].

Importantly, the neutralizing potency of the plasma samples did not correlate with the IgG antibody responses to linear epitopes.

### 3.3. Immune Responses against ST

The plasma samples from all the enrolled ETEC-infected patients were assessed for IgG antibodies against ST at acute, early, and late convalescence phases (within 2–3 days, day 7, and day 30 after presenting with symptoms, respectively) (Figure 9). Of the 37 adults and 27 children who presented with a ST-positive ETEC infection, only one adult (2.7%) and three children (11%) showed an at least 2-fold increase in plasma anti-ST IgG on day 7 or day 30 compared to day 2 after presenting with acute ETEC infection. Only one child had a more than 4-fold increase in plasma anti-ST IgG.

Plasma from these four subjects was assessed in an ST neutralization assay based on the ST-induced cGMP induction in T84 human colon epithelial cells. No significant increase in the ST-neutralizing capacity of plasma collected at day 7 or 30 compared to day 2 plasma of the same subject could be detected in any of the analyzed samples. This suggests that the anti-ST IgG antibodies present in the plasma after natural ST+ ETEC infection were not sufficient to achieve neutralization of ST under the experimental conditions tested.

The plasma from 40 healthy adults from Bangladesh was also collected within this study. These samples were compared for the anti-ST IgG to those from an endemic sample collection of 401 healthy adults and a non-endemic collection of 49 sera (Figure 10). While the geometric mean ST neutralization signal was double in the endemic samples compared to that obtained with the non-endemic samples, there were only a few individuals identified with high levels of anti-ST IgG. Those samples with the highest ST titers were tested for the ST-neutralizing capacity, but none could reduce the ST-induced cGMP induction in T84 cells.

## 4. Discussion

ETEC infection is an important cause of diarrhea in children in low- and middle-income countries (LMIC) and is also one of the main causes of traveler’s diarrhea (TD). The vaccine development efforts against ETEC have been mainly focused on the B-subunit of the heat-labile toxin (LTB) and surface colonization factors [16].

In the study reported here, we found that 42% of the enrolled children were infected with an LT-only containing ETEC strain. In contrast, only 14% of the adults were infected with an ETEC strain that expressed only LT (LT+ST−). It can be speculated that adults have gained protective immunity against LT through multiple previous ETEC infections, hence less adults were getting sick from LT-only-expressing ETEC or producing milder symptoms which did not need to be treated in a medical center. In contrast, ST, being a poor antigen, did not induce sufficient protective immunity, leaving even adults vulnerable to ST-expressing ETEC. This is supported by data from multiple studies. The contribution of LT+ or ST+ ETEC to diarrheal stools has been shown to be of a similar proportion in children in their first and second year of life [17]. Another study showed that adult travelers to endemic regions were infected at a rate of 44% with LT+ST− ETEC strains [18]. Thus, in children in endemic countries and adults from non-endemic settings, LT-only-containing ETEC infections occur at a similar rate as those with ST-containing ETEC strains. However, ST-expressing ETEC has been shown to be associated with an increased risk of death in infants aged 0–11 month [1] and an increased severity of disease. While LT- ETEC could be found at similar proportions in diarrheal and non-diarrheal surveillance stools in children in their first and second year of life, ST+ ETEC strains were more prevalent in diarrheal stools [17]. Traveler’s diarrhea (TD) among international travelers has been associated with ST-containing, but not LT-containing ETEC [19], and more severe symptoms of TD are associated with an STh-containing ETEC infection [18]. The induction of anti-ST immunity could be critical for protection against ETEC.

It has been previously shown that LT+ ETEC infection conferred a 45% protection against symptomatic infection with LT+ ETEC [20]. In this study, we detected lower responder rates against LT in both children and adults after natural infection with LT+ ETEC. It is possible that our study population included subjects who were only colonized by ETEC but suffering from diarrhea caused other pathogens which were not tested for. It could also be considered that in endemic areas, only those adults who have poor immune responses develop diarrhea after ETEC infection, biassing our study towards the selected vulnerable subjects. Alternatively, the plasma antibody levels may not fully reflect mucosal immune responses where the toxin-neutralizing antibodies exert their protective effects.

We found that both LT subunits were immunogenic and contributed to the neutralization activity. Interestingly, the anti-LTB IgG and IgA responses correlated in the children but not in the adults where we detected higher IgA levels even when the IgG levels were low.

The anti-LTB IgG titers strongly correlated with the neutralizing potency in children and adults. This confirms results from oral vaccination with the LTB-containing vaccine ShigETEC, for which LTB IgG titer also correlated with neutralizing potential [21]. Due to its homology with CTB, it was not surprising to observe a strong correlation with CTB neutralizing capacity, confirming the potential cross-protection that LTB-based vaccination strategies could confer against cholera.

The linear epitope mapping identified, for the most part, relatively weak antibody responses to LTB, which was inconsistent across different individuals. The epitope responses to LTA were stronger, and there were two particular peptides (positions 54–66 and 138–150) that elicited a response in almost all adults and children. These two peptides also overlapped with previously reported epitopes [15]. A stretch in the C-terminal end (positions 206–242) was also recognized by the antibodies from most of the subjects. There was, however, a lack of correlation between the antibodies to linear epitopes and the neutralizing activity of each sample. This indicates that LT-neutralizing antibodies primarily bind conformational epitopes.

In this study we found weak responses against ST after ST+ ETEC infection. While this could be due to the relatively small number of individuals included in the study, in general, the background level of anti-ST antibodies in endemic and nonendemic populations were not substantially different. Only a few plasma samples were identified with increased anti-ST IgG levels, but none were high enough to show neutralizing activity. This is supported by a previous observation that natural ETEC infections do not appear to induce a meaningful immune response against ST, which is presumably due to the small size of ST [22]. The poor immunogenicity of ST could also be linked to its structural homology to the endogenous hormones guanylin and uroguanylin. However, it is possible to induce neutralizing antibodies against ST in animals. This was achieved by vaccination with large amounts of the ST protein in order to induce high serum antibody levels, which did show ST-neutralizing activity [12,23,24]. In a previous human vaccination study with synthetic ST fused to an LTB epitope, a total of 60 mg of protein orally applied in three or four doses of 20 or 15 mg each, respectively, were necessary to induce the anti-ST antibodies with neutralizing capacity [25]. Such levels were not achieved in humans after natural infection since we did not find any plasma with ST-neutralizing activity.

The improvement of the current vaccine development efforts to include large amounts of ST protein could provide a tool to tackle the important contribution of ST-containing ETEC strains to severe disease and death among children.

## Figures and Tables

**Figure 1 microorganisms-11-02524-f001:**
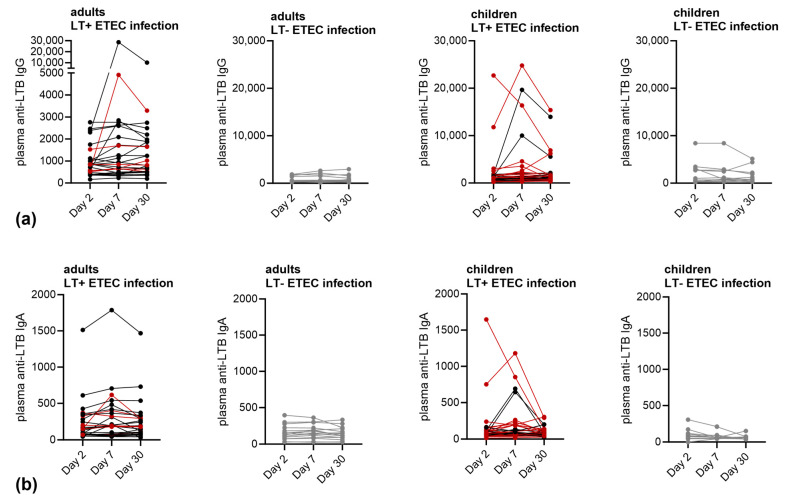
Anti-LTB plasma antibody levels in ETEC-infected adults and children. Plasma anti-LTB (**a**) IgG or (**b**) IgA endpoint titers of LT-positive (red = LT+ST− ETEC infection, black = LT+ST+ ETEC infection), or LT-negative ETEC (grey) infected adults or children at day 2, day, 7, and day 30 post hospitalization.

**Figure 2 microorganisms-11-02524-f002:**
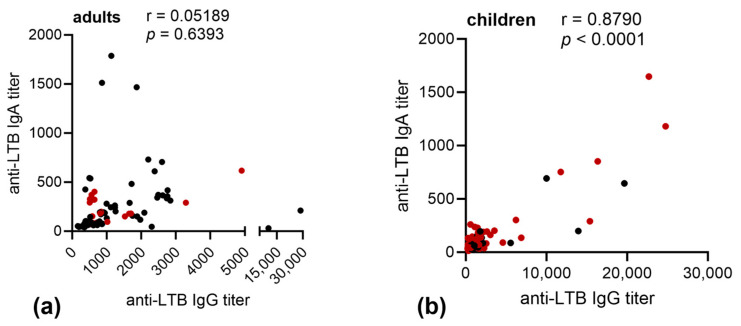
Correlation of anti-LTB IgG and IgA titers in plasma of patients infected with LT-positive ETEC strains. Pearson’s correlation of endpoint titers of plasma anti-LTB IgG and of IgA of (**a**) adults or (**b**) children with confirmed LT-positive ETEC infection (red = LT+ST− ETEC infection, black = LT+ST+ ETEC infection) at day 2, day 7, and day 30 after hospitalization.

**Figure 3 microorganisms-11-02524-f003:**
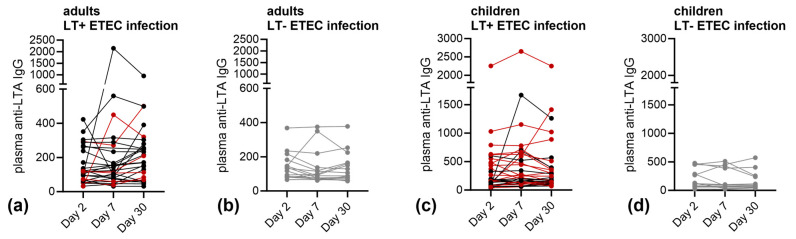
Anti-LTA IgG antibody responses in plasma in ETEC-infected adults and children. Plasma anti-LTA IgG endpoint titers of (**a**,**c**) LT-positive (red = LT+ST− ETEC infection, black = LT+ST+ ETEC infection) or (**b**,**d**) LT-negative (grey) ETEC infected (**a**,**c**) adults or (**b**,**d**) children at days 2, 7 and 30 post hospitalization.

**Figure 4 microorganisms-11-02524-f004:**
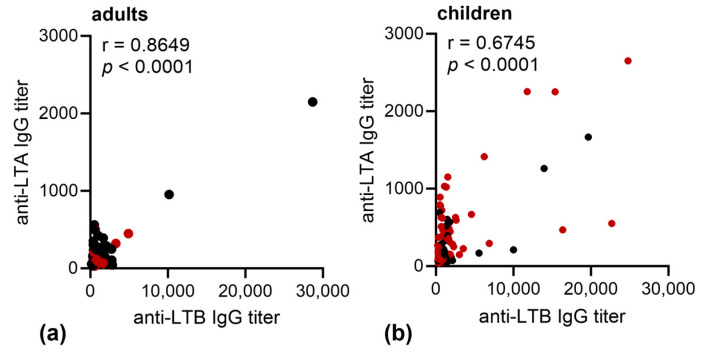
Correlation of anti-LTB and anti-LTA IgG titers in plasma of patients infected with LT-positive ETEC strains. Pearson’s correlation of endpoint titers of plasma anti-LTB IgG and of anti-LTA IgG of (**a**) adults or (**b**) children with confirmed LT-positive ETEC infection (red = LT+ST− ETEC infection, black = LT+ST+ ETEC infection) at day 2, day 7, and day 30 after hospitalization.

**Figure 5 microorganisms-11-02524-f005:**
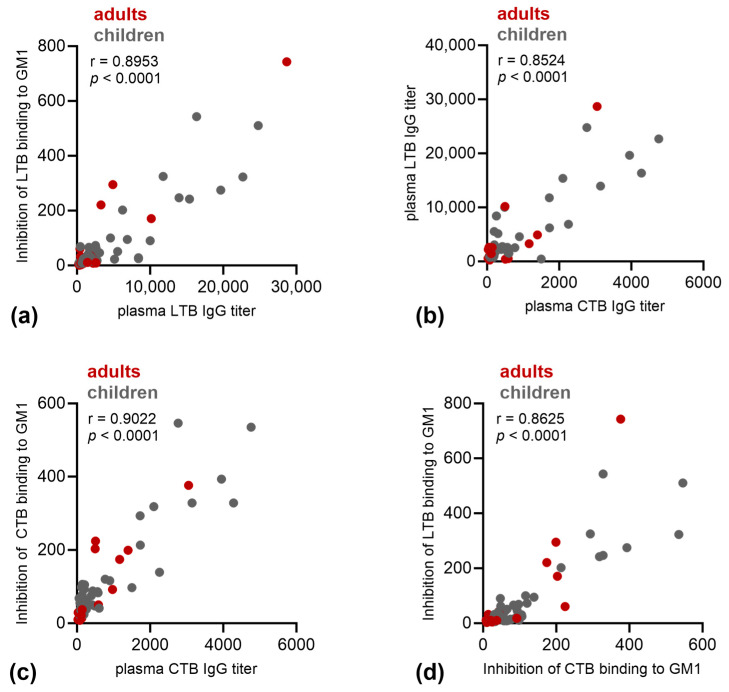
Correlation of toxin neutralization capacity and anti-toxin IgG titers. Plasma of adults (red) and children (grey) was assessed for (**a**,**d**) anti-LTB IgG or (**b**,**c**) anti-CTB IgG and neutralization potential of (**a**,**d**) LTB or (**c**,**d**) CTB measured by blocking of binding of LTB or CTB to GM1, expressed as DF50. Pearson’s correlation was performed, and r and *p* values are given.

**Figure 6 microorganisms-11-02524-f006:**
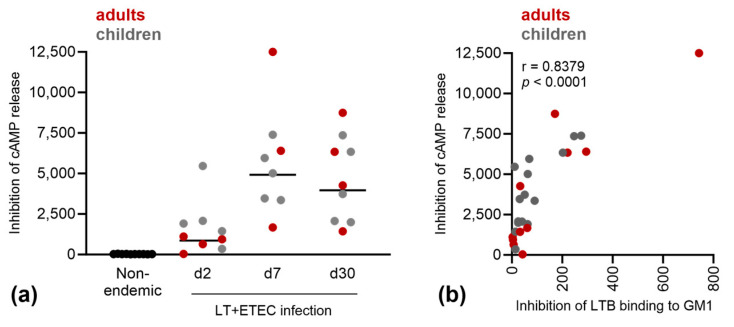
LT neutralization activity. (**a**) LT neutralization activity of serum/plasma from non-endemic healthy adults (black) or endemic healthy adults (red) or endemic children (grey) was determined by blocking LT-mediated cAMP release in T84 cells. Inhibition of cAMP release was calculated by multiplying the percentage of neutralization of a fixed amount of LT by the dilution factor of the plasma used. (**b**) Pearson’s correlation of LT neutralization activity, measured as blocking of LT induced cAMP release from T84cells, and LTB neutralization, measured as blocking of binding of LTB to GM1, expressed as DF50.

**Figure 7 microorganisms-11-02524-f007:**
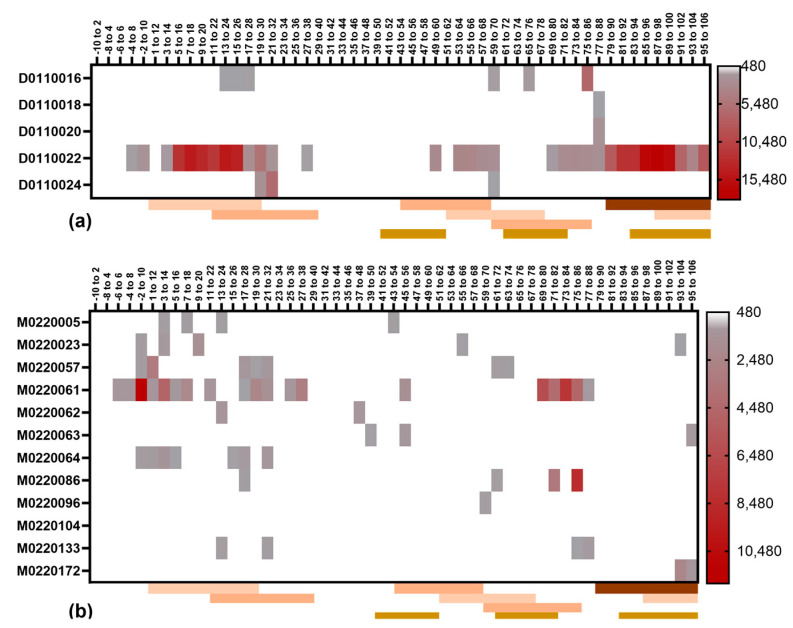
LTB linear epitope mapping using plasma samples from patients with LT+ ETEC infection. Heat maps are generated from peptide array data for LTB IgG for selected plasma samples (each row represents one individual) from (**a**) adult or (**b**) paediatric patients. Consensus LTB sequence based on the data set from all five LTB variants was used and the highest value obtained for each peptide variant was used for each sample. Amino acid positions for each peptide in LTB consensus sequence are given at the top of each heatmap. Cut off for positive signal was set to be 480. Published linear epitopes are marked below each plot in orange [13] or brown [14].

**Figure 8 microorganisms-11-02524-f008:**
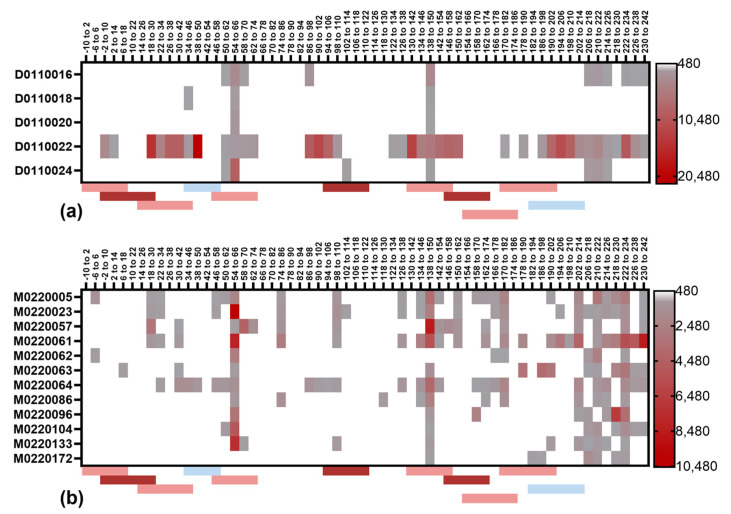
LTA linear epitope mapping using plasma samples from patients with LT+ ETEC infection. Heat maps are generated from peptide array data for LTA IgG for selected plasma samples (each row represents one individual) from (**a**) adult or (**b**) paediatric patients. Consensus LTA sequence based on the data set from all five LTA variants was used where the highest value obtained for each peptide variant was used for each sample. Amino acid positions for each peptide in LTA consensus sequence are given at the top of each heatmap. Colour scale for each map is given on the right side, cut off for positive signal was set to be 480. Published linear epitopes are marked below each plot in red (neutralizing) or blue (non-neutralizing) epitopes. Epitopes marked with dark red have strong neutralization capacity [15].

**Figure 9 microorganisms-11-02524-f009:**
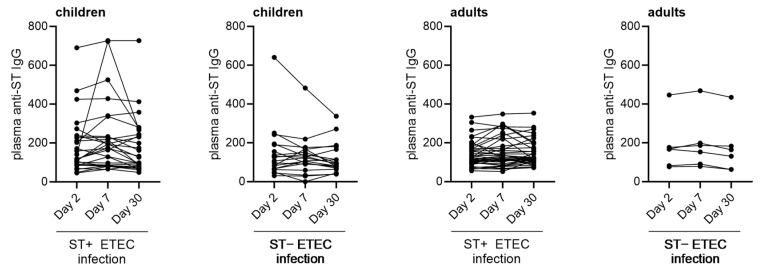
Anti-ST IgG antibody responses in plasma of ETEC-infected children and adults. Endpoint anti-ST IgG titers of children or adults infected with ST-positive or ST-negative ETEC at days 2, 7, and 30 post hospitalization.

**Figure 10 microorganisms-11-02524-f010:**
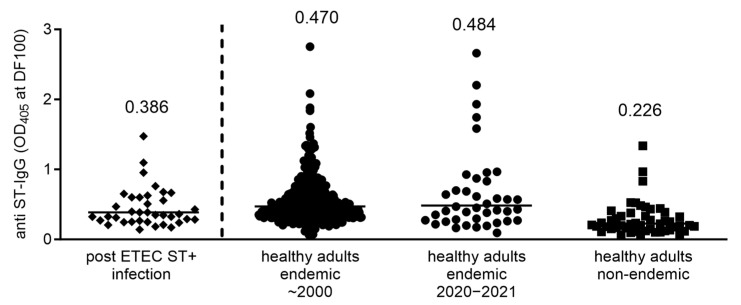
Anti-ST IgG antibody responses in plasma or serum of healthy endemic or non-endemic populations, as well as of adult patients infected with ST positive ETEC strains. Data are presented as OD_405nm_ values at 1:100 dilution of serum or plasma used. Geometric mean of the values for each population are presented. For infected patients, plasma samples taken on day 30 after admission were used.

**Table 1 microorganisms-11-02524-t001:** Gender and age distribution in enrolled ETEC-infected adults and children.

	Gender		Age			
	Male Number (%)	Female Number (%)	1–11 Months Number (%)	12–23 Months Number (%)	2–5 Years Number (%)	18–45 Years Number (%)
**Children** (*n* = 46)	26	20	5	14	27	
	(56.5%)	(43.5%)	(10.9%)	(30.4%)	(58.7%)	
**Adults** (*n* = 43)	17	26				43
	(39.5%)	(60.5%)				(100%)

**Table 2 microorganisms-11-02524-t002:** ETEC infection typing based on the presence of the LT and ST genes.

	Infection				
	LT+/ST− Number (%)	LT+/STh+ Number (%)	LT+/STp+ Number (%)	STh+ Number (%)	STp+ Number (%)
**Children** (*n* = 46)	19	9	6	7	5
	(41.3%)	(19.6%)	(13%)	(15.2%)	(10.9%)
**Adults** (*n* = 43)	6	21	1	10	5
	(14%)	(48.8%)	(2.3%)	(23.3%)	(11.6%)

## Data Availability

No new data other than the published data were created.

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
