# Peer review of "Anti-Toxin Responses to Natural Enterotoxigenic Escherichia coli (ETEC) Infection in Adults and Children in Bangladesh"

_microorganisms, 2023, doi:10.3390/microorganisms11102524_

Round 1

Reviewer 1 Report

-       -        This study aimed to characterize the anti-toxin antibody profiles in plasma samples of infected adults and children living in Bangladesh, endemic for ETEC infections.  However, the study is properly designed and the materials address the purpose of the study, there are some minor comments that should be modified.

  Minor comments

-        Please revise the manuscript for English language and carefully correct sentences related mistakes and grammatical errors.

-        In the title: Please write the full name of ETEC

-        Line 60: rewrite as: adenylate cyclase resulting in increasing  the  intracellular …….

-        Line 84: For screening of diarrheal patients (delete of)

-        Line 178” This urban research area has been used  for field: replace used for  with subjected to

-        Line 184: A total of 1097

-        Line 185: replace 130 with One hundred- thirty

-        Line 338: write full name of LMIC

-        Line 342:  we found that 42% of children enrolled in the study were (replace this sentence with :  we found that 42% of the enrolled children were).

 -        I suggest that the manuscript is suitable for publication after minor revision.

-        This study aimed to characterize the anti-toxin antibody profiles in plasma samples of infected adults and children living in Bangladesh, endemic for ETEC infections.  However, the study is properly designed and the materials address the purpose of the study, there are some minor comments that should be modified.

  Minor comments

-        Please revise the manuscript for English language and carefully correct sentences related mistakes and grammatical errors.

-        In the title: Please write the full name of ETEC

-        Line 60: rewrite as: adenylate cyclase resulting in increasing  the  intracellular …….

-        Line 84: For screening of diarrheal patients (delete of)

-        Line 178” This urban research area has been used  for field: replace used for  with subjected to

-        Line 184: A total of 1097

-        Line 185: replace 130 with One hundred- thirty

-        Line 338: write full name of LMIC

-        Line 342:  we found that 42% of children enrolled in the study were (replace this sentence with :  we found that 42% of the enrolled children were).

 -        I suggest that the manuscript is suitable for publication after minor revision.

Reviewer 2 Report

Enterotoxigenic Escherichia coli (ETEC) is an enteric pathogen that causes substantial mortality and morbidity in low to middle income countries. ETEC produces the heat-stable (ST) and heat-labile (LT) enterotoxins, and it is believed that anti-toxin (anti-ST and anti-LT) immunity would suffice to protect against all ETEC diarrhea. This manuscript by Girardi et al titled “Anti-toxin responses to natural ETEC infection in adults and children in Bangladesh” should be of general interest to those readers of Microorganisms, especially those interested in enteric infections in endemic settings.  However, the manuscript could be improved for clarity in the data presentation and overall story. Here, the authors designed a study to evaluate anti-toxin immune responses following natural infection in children and adults in an endemic setting in Bangladesh. Patients were enrolled upon being given informed consent while seeking medical intervention for secretory diarrheal disease.  The authors then screened these patients for the presence of ETEC. Then, serum anti-toxin (anti-LT and anti-ST) responses were measured during acute, early, and late convalescence phases.  Then authors used neutralization assays to confirm anti-toxin responses in vitro. Further the authors used linear epitope mapping to screen for potentially protective LT epitopes recognized by antisera.

Majors:

Line 80, Please explain how plasma toxin-neutralizing potency is an indicator of mucosal immunity. 

Line 91-92, Why did the inclusion criterion 1 state positive stool culture for Escherichia coli or Shigella? Were any Shigella identified via PCR primers? Do some of the data represent Shigella hybrids carrying ETEC toxins? Or does this mean that some of the people were co-infected with ETEC and Shigella?

Were any metrics on diarrheal volume or frequency measured?  This would help set the stage for the anti-toxin data.

Line 134, Most assays neutralize 2 ng ST toxin in the T84 assay. Neutralization of 5 ng ST toxin is not trivial and may be a reason for lackluster ST neutralization.  Please comment on this.

Line 205 and supplemental table, please explain in the text why the IgG responses are calibrated to day 2 and IgA responses are calibrated to day 30 post-infection.

Line 219 AND line 237, please include the red/black labeling so that the readers can understand how the presence of ST affects the development of anti-LT responses in correlations (and since Figs 1 and 3 have it graphed that way). Also consider reordering the phrases regarding adults and children in line 217-218 since you show adult data in Fig 2A and children data in Fig 2B.

Line 242, there needs to be some description as to how samples were selected. Were all 3 samples (d2, d7, d30) from a positive responder were assessed? If so, how many different patients are shown? Again if so, can samples from the same patient(s) (d2, d7, d30) be grouped so the readers can understand if the correlations are general or being more heavily weighted by a particular individual(s)?

Line 248, the authors never show the data for anti-CTB responses used in the correlations.  Possibly include the data as supplement. It would also be important to know if the same individuals were the high responders for anti-CTB and anti-LTB titers

Line 262-265 and figure 6.  What is inhibition of cAMP release mean? Percent? It is unclear how the authors used cAMP readout and converted it to a inhibition of cAMP release measurement (y-axis).  Please be more straight-forward in the data presentation to show the cAMP (pmol/ml) data at a fixed LT mass with and without neutralizations.

Figure 7/8.  It is not clear from the text what these figures add that was not known previously.  

Discussion: It should be noted somewhere that anti-ST immunity could be critical for ETEC protection. Line 400-402, the cited study used 15 mg ST-LTB conjugate over 4 doses or 20 mg ST-LTB conjugate over 3 doses. Each indeed totals 60mg, but the way that this is stated makes it seem like 60 mg was used in each of the 3 or 4 doses.  Please be clearer in this wording.  This same study demonstrated that there was a substantially higher amount of anti-ST mucosal IgA (1:512 titer) than serum IgG (1:16 titer). 

Minors: lines 37-40 and lines 42-44 describe similar problems

Line 43, specify what is meant by ‘endocrine delays’ and provide a specific citation

Line 52, the phrase “and results in water retention in the gut” is misleading.  The toxins ultimately cause cellular export of chloride ions that cause ion imbalance and water flows from the cells into the lumen of the bowel via osmosis.  Therefore, water is not being retained in the gut.

Line 59, LT-holotoxin is internalized, not just LTA.

Line 63-64. More than LTB is considered as a mucosal adjuvant. LT, dmLT, LTB, and LTA.

Line 66: specify that cross-protection from Dukoral are likely LT-ETEC infections.

line 69, strike to epithelial cells and induces cell activation via binding

line 128, LT beta is not used. 

Line 266.  Fig 6 a does not show that neutralizing activity increased over time.  The neutralizing activity increased between d2 and d7.  Then the neutralizing activity decreased between d7 and d30. This is a confusing figure.

Line 281-282 include citations in line

Line 349, cite those studies

Line 365, who were NOT only colonized by ETEC, add the word not

English is fine
